# Transient Elastography in Community Alcohol Services: Can It Detect Significant Liver Disease and Impact Drinking Behaviour?

**DOI:** 10.3390/biomedicines10020477

**Published:** 2022-02-17

**Authors:** Mohsan Subhani, David J. Harman, Robert A. Scott, Lucy Bennett, Emilie A. Wilkes, Martin W. James, Guruprasad P. Aithal, Stephen D. Ryder, Indra Neil Guha

**Affiliations:** 1Nottingham Digestive Diseases Biomedical Research Centre (NDDC), School of Medicine, University of Nottingham, Nottingham NG7 2UH, UK; david.harman13@gmail.com (D.J.H.); mszrs3@exmail.nottingham.ac.uk (R.A.S.); lucy.bennett@nottingham.ac.uk (L.B.); emilie.wilkes@nuh.nhs.uk (E.A.W.); martin.james@nuh.nhs.uk (M.W.J.); guru.aithal@nottingham.ac.uk (G.P.A.); stephen.ryder@nuh.nhs.uk (S.D.R.); neil.guha@nottingham.ac.uk (I.N.G.); 2NIHR Nottingham Biomedical Research Centre, Nottingham University Hospitals NHS Trust and the University of Nottingham, Nottingham NG7 2UH, UK

**Keywords:** cirrhosis, ArLD, alcohol abuse, transient elastography, diagnosis, prospective

## Abstract

**Introduction**: Alcohol is the leading cause of cirrhosis in Western populations. The early identification of high-risk drinkers followed by intervention is an effective way to reduce harm. We aim to assess the feasibility of integrating transient elastography (TE) into community alcohol services, and to determine its impact on modifying drinking behaviours. **Method**: A prospective cohort study was conducted at a community alcohol clinic in Nottingham, UK (April 2012 to March 2014). Patients (>18 years) with a primary alcohol problem were recruited. Those known to liver services or those known to have chronic liver disease were excluded. Significant liver fibrosis was defined by a liver stiffness of >8 kilopascal (kPa). Follow-up was for a minimum of six months. Data were descriptively analysed for significant differences between patients with a normal liver stiffness versus raised liver stiffness. **Results**: 156 patients were invited; *n* = 87 attended and *n* = 86 underwent successful TE. The majority were male (*n* = 53, 70.0%), and the mean age was 46.3 years (SD ± 9.8). Median liver stiffness was 6.9 kPa (range 3.1–75.0kPa). Clinically significant liver fibrosis was identified in *n* = 33 (38.4%), of which *n* = 6 were in the cirrhotic range (≥15 kPa). The baseline median self-reported alcohol intake for normal stiffness was 126 units per week (range 24–378) and in raised stiffness was 149.0 units per week (range 39.0–420.0); this difference was nonsignificant (*p* = 0.338). The median reduction in self-reported alcohol intake in the whole cohort was 65.0 units per week (range 27.0–88.0, *p* < 0.001); in the normal liver stiffness group it was 25.0 units per week (range 18.0–75.0, *p* = 0.154), and in the raised liver stiffness group it was 78.5 units per week (range 36.0–126.0, *p* < 0.001). **Conclusion**: The study demonstrated that transient elastography is a feasible tool to stratify clinically significant liver disease in community alcohol services. It can stimulate a change in high-risk drinking behaviour and a normal liver stiffness result does not provide false reassurance to participants.

## 1. Introduction

Alcohol is the leading cause of cirrhosis in most high-income countries [1,2,3]. Over the last three decades, the UK has observed a 400% rise in liver-disease-related mortality, and in 2020, Public Health England reported that alcohol-specific deaths had reached their highest level since 2001 [1,4,5]. The estimated cost to the National Health Services (NHS) to treat alcohol-related problems is over GBP 3.5 billion annually [2,3]. According to the World Health Organisation (WHO) 2018 report, globally alcohol use contributes to over 3 million annual deaths, 132.6 million disability-adjusted life years lost (DALYs), and over 200 medical conditions [6]. Nationally in the UK, Nottingham is among the worst for alcohol dependence, alcohol-related hospital admission, and mortality [7,8].

Alcohol-related liver disease (ArLD) has a positive dose response relationship with the amount of alcohol consumed; the risk starts from an alcohol consumption as low as 14–72 g per week [9,10]. As compared to long-term abstainers, the consumption of one drink a day (12 g of pure alcohol) significantly increases the risk of cirrhosis in women (relative risk 1.64, 95% CI 1.07, 2.51). For an alcohol consumption of five to six drinks per day, the relative risk (RR) of cirrhosis in both men and women is 6.26 (95% CI 2.38, 16.50), and for more than seven drinks a day it is 10.70 (95% CI 2.95, 38.78) [10]. Liver disease causes no symptoms in its earlier stages; over half of patients have liver disease first diagnosed after an emergency hospital admission at a stage when the scope of any medical or behavioural intervention is minimal [1,2,11]. ArLD is 12 times more likely to present late compared to other aetiologies of liver disease [12]. Once patients with alcohol misuse develop cirrhosis, the prognosis becomes exceptionally poor; the mortality rate for alcohol-related cirrhosis has been reported as high as 75% at 5 years and 91% at 15 years [13].

A recent study showed that people who died of ArLD had a median of 25 hospital attendances in the five years before their death [14]. Each of these interactions with healthcare services presents a missed opportunity [15]. The early identification of high-risk patients followed by intervention is an effective way to reduce harm [1,16]. Current screening strategies to detect clinically significant liver fibrosis rely on serum liver function blood tests (LFTs), which are known to have a poor sensitivity. The yield of significant liver disease following the investigation of abnormal LFTs in the community is less than 3% [17]. A number of noninvasive tests (NITs) such as transient elastography (TE) and enhanced liver fibrosis (ELF) tests are now available, which can reliably test for the presence or absence of significant liver fibrosis [18]. The use of these tests is not widely embedded in high-risk community services, including alcohol services [2,16].

Alcohol brief interventions do impact drinking behaviour and reduce alcohol consumption [19,20]. Despite several behavioural interventions for alcohol misuse having been in clinical practice for over two decades, alcohol-related harm is on an alarming rise [1]. The early diagnosis of liver fibrosis provides an opportunity to intervene and reduce or stop alcohol intake. This is known to be the most effective way of preventing liver disease progression [21]. Providing feedback to patients based on NITs of liver disease can impact drinking behaviour [22,23]. It has been hypothesised that the demonstration of the degree of liver damage related to excess drinking, combined with advice to reduce alcohol consumption, would be more effective [23]. In contrast, concerns have also been raised regarding the risks of NIT-based feedback methods potentially providing false reassurance, leading to unintended negative consequences such as exacerbating pre-existing addictive behaviours [24]. We aimed to assess the feasibility of integrating transient elastography into community alcohol services, to assess the effectiveness of TE in identifying significant liver disease in high-risk asymptomatic individuals, and to determine its impact on modifying drinking behaviours.

## 2. Methods

This prospective cohort study was conducted at Oxford Corner Community Alcohol Services in Nottingham, United Kingdom. Most community alcohol services in the UK are run by non-NHS providers. These services can vary by locality, but most provide structured community-based interventions such as psychological and recovery support with or without medical therapy [25]. The standard treatment varies based on individual patient needs. In general, the pharmacological interventions include management of alcohol withdrawal and prevention of alcohol relapse. Examples of the most common medications used in these services to prevent relapse are Disulfiram, Naltrexone, and Acamprosate. The psychological interventions comprise a subgroup of interventions such as motivational intervention, family and social network intervention, cognitive-behaviour-based relapse prevention, and behavioural self-control. The majority of patients self-present to these services, but often general practitioners (GPs) or other healthcare providers can directly refer or signpost patients to this service.

Data collection was carried out as part of a prospective study to assess the feasibility of a novel diagnostic algorithm targeting patients with risk factors for chronic liver disease in a primary care setting. The project was started as part of service evaluation process after local regulatory approval (10/HO405/8), subsequent full ethical approval was obtained from East Midlands—Leicester Central Research Ethics Committee (13/EM/0123) on 10 April 2013. Informed consent was obtained from all subjects involved in the study. Consecutive patients (>18 years) who presented to alcohol services with a primary alcohol problem between April 2012 and March 2014 were invited to participate in the study. Patients who had a primary substance misuse other than alcohol, had an established diagnosis of chronic liver disease, had undergone a TE previously, or were known to liver services were excluded.

Study-related information was given to patients at index presentation. Patients were invited to attend a baseline study appointment to have a TE using a portable Fibroscan with a FS402 probe (Echosens, Paris, France) and blood tests. TE was performed by one of three experienced, trained nurses, all of whom had performed more than 50 examinations as a part of the TE service prior to the start of study [26]. All investigations and scans were performed at Oxford Corner. Patients were followed up for a minimum of six months after TE.

Liver stiffness measure (LSM) of less than 8 kilopascals (kPa) was reported as normal liver stiffness, and LSM of >8 kPa as raised liver stiffness, which indicated possible clinically significant hepatic fibrosis [27]. Patients with a liver stiffness of >8 kPa were reviewed by a consultant hepatologist within the community. Baseline patient characteristics (age and sex) and self-reported alcohol intake before TE were noted from Community Alcohol Services patient notes. A follow up self-reported alcohol intake was retrieved from medical records after six months. No repeat blood tests were carried out as part of the study.

Nottingham University Hospital (NUH) laboratory test reference guide was used to define normal reference ranges for blood results. The normal reference ranges for the tests were as follows: alanine aminotransferase (ALT) 0–45 U/L, aspartate aminotransferase (AST) 0–35 U/L, gamma glutamyl transferase (GGT) 0–55 U/L.

The normally distributed variables were expressed as mean ± standard deviation (SD), non-normally distributed variables as median with range, and categorical variables as frequency. Descriptive analysis was performed to describe the distribution of key variables among study population. Diagnostic rate for significant liver stiffness and change in self-reported alcohol intake were used as primary outcome. The correlation between continuous normally distributed variables was assessed by parametric tests (Pearson’s correlation coefficient, unpaired T-test, ANOVA test) and non-normally distributed by nonparametric tests (Spearman’s correlation coefficient, Mann–Whitney U test). Depending on sample size, categorical variables were analysed by the chi-squared test or using Fisher’s exact test, with results reported as number (percentage).

Statistical analysis was conducted using Statistical Package for the Social Sciences (SPSS version 26.0, IBM corporation, Armonk, NY, USA), and Prisma GraphPad (version 8.0, San Diego, CA, USA). STROBE reporting guidelines for reporting observational studies in epidemiology are used throughout this article.

## 3. Results

A total of *n* = 156 patients who consecutively presented to Oxford Corner Alcohol Services were invited to participate in the study. Eighty-seven patients (55.7%) attended a subsequent study appointment, of which 86 patients had a successful estimation of liver stiffness by TE, and 85 patients had blood tests carried out (Figure 1). The majority were male (*n* = 53, 70.0%) and the mean age was 46.3 years (SD ± 9.8). The median self-reported alcohol intake was 145 units per week (range 24–240). The median liver stiffness in the whole cohort was 6.9kPa (range 3.1–75.0kPa). The mean ALT was 64.5 U/L (SD ± 52.5, normal reference range 1–45 U/L), and the mean GGT was 568.6 U/L (SD ± 757.4, normal reference range 0–55 U/L) (Table 1). The AST:ALT ratio was >1.0 in *n* = 64 (74.4%) of patients. Overall, 53 patients (61.6%) had normal liver stiffness and *n* = 33 (38.4%) had a significantly elevated liver stiffness. Of those with a raised liver stiffness, six (6.8%) had liver stiffness in the cirrhotic range (≥15 kPa).

On comparison of the patients who had normal liver stiffness versus the patients with raised liver stiffness, there was no significant difference in self-reported alcohol intake at baseline (*p* = 0.338). The median reduction in self-reported alcohol intake after six months compared to baseline in the whole cohort was 65.0 units per week (range 27.0–88.0, *p* < 0.001). For those who had normal liver stiffness, the reduction in alcohol consumption was 25.0 units per week (range 18.0–75.0), which was statistically nonsignificant (*p* = 0.154). In contrast, those with raised liver stiffness reduced their alcohol intake by a median of 78.5 units per week (range 36.0–126.0), which was statistically significant (*p* < 0.001). At six months, 19.0% (*n* = 4) of the normal liver stiffness group were abstinent compared to 26% (*n* = 7) in the raised liver stiffness group (*p* = 0.73).

In the raised liver stiffness group, follow-up data on self-reported alcohol intake were available for 26 patients, of which 17 patients were reported to have reduced their alcohol consumption and 1 patient increased their consumption. In the group with normal liver stiffness, follow-up data on self-reported alcohol intake were available for 21 patients, of which 14 patients reduced their alcohol consumption and 5 patients increased their alcohol consumption. There was no significant difference between the two groups in the proportion of individuals who reduced (*p* = 1.00) or increased (*p* = 0.08) their alcohol intake (Table 2).

ALT was available in *n* = 85 (98.8%) of patients. ALT was normal in 23 patients (43.4%) with normal stiffness and in 8 patients (25%) with raised stiffness (Table 2). The mean ALT at baseline was significantly lower in the normal stiffness group compared to the raised liver stiffness group (53.0, SD ± 43.4 and 83.1 SD ± 60.8, *p* = 0.01) (Table 1).

Gamma-glutamyl transferase (GGT) was available in *n* = 59 (68.6%) of patients. GGT was normal in 15.0% (*n* = 5) of those with normal stiffness and 4.0% (*n* = 1) of those with raised stiffness (Table 2). The mean GGT at baseline was significantly lower in the normal stiffness group compared to the raised liver stiffness group (226.7, SD ± 260.8 and 1033.6, SD ± 949.7, *p* < 0.001) (Table 1).

## 4. Discussion

The study demonstrates that the integration of noninvasive liver stiffness measurement by TE into community outreach alcohol services is feasible. More than half of the invited patients attending community alcohol services took part in TE-based evaluation, an uptake similar to that previously demonstrated in primary-care-based studies screening for chronic liver disease [28,29]. The major potential benefit for patients undergoing TE in community alcohol services is an easy-to-access noninvasive test for significant liver fibrosis. Four in ten asymptomatic individuals screened in these services showed evidence of significant liver fibrosis, and worryingly, one in ten had cirrhosis. This highlights the community burden of undiagnosed liver disease, especially in a high-risk population, and the need for the implementation of the National Institute of Clinical Excellence (NICE) guidelines, which state that adults with high levels of alcohol dependency should be assessed [25]. These guidelines recommend that these patients should be offered intensive, structured, community-based interventions (with or without medical therapy), as these provide the best chance of achieving and maintaining abstinence from alcohol [25]. Our study raises the possibility that the use of TE in these settings may enhance the effectiveness of behavioural interventions.

At the six-month follow-up, the group with raised liver stiffness reduced their alcohol intake by a median of 75.0 units per week compared to 25.0 units per week in the normal liver stiffness group. This may imply an added behavioural impact of receiving feedback based on NITs of liver disease. Reassuringly, a normal liver stiffness result did not provide false reassurance to participants with no evidence of significant increased consumption or a lack of reduction in those with a normal TE value. Almost a quarter of participants with raised liver stiffness had normal ALT. The lack of a correlation between ALT and the degree of liver fibrosis has been well recognised. Supplementing liver function tests with NITs to stratify the severity of liver disease in high-risk areas such as community alcohol services can significantly enhance the early-detection rate [22,30,31]. Similar changes in alcohol consumption in response to NITs of liver disease have been previously reported in patients with hazardous alcohol consumption in primary and secondary care but not in community alcohol services [22,23,32]. This change in alcohol consumption is backed up by previous evidence showing that providing personalised health care communications enhances the motivation to overcome addictive behaviour [33,34]. For hazardous and harmful alcohol users, providing feedback based on a simple liver fibrosis test prompts a reduction in alcohol consumption for those both with and without evidence of liver damage [22].

The current study also demonstrates that TE is an acceptable intervention for patients presenting to community alcohol services, which is consistent with previously reported evidence exploring the acceptability of TE to screen for CLD in a UK primary-care setting [28]. An attendance rate of 56% for TE appointments is in keeping with other community screening algorithms in the UK in which patients attend tests in the community, such as the NHS colorectal cancer screening programme [32] and the Southampton Traffic Light screening test of patients with hazardous alcohol consumption [22]. A more targeted outreach screening programme of high prevalence liver disease populations, such as people who inject drugs, incorporating transient elastography in Denmark had a remarkably similar uptake [31]. A recent study showed that establishing community-based point-of-care services to manage chronic liver disease (CLD) in people who are homeless (PWAH) significantly improved the identification of clinically significant liver disease and compliance with medical treatment [35]. A prospective community-based observational study involving harmful drinkers demonstrated that performing TE enhances subsequent engagement in secondary-care liver services [36].

Self-motivation has been widely shown to be an independent factor in behavioural change, and self-motivated people are more likely to sustain long-term recovery from substance misuse [37]. Self-presentation was the most common source of referral, and as per the transtheoretical model of health behaviour change, our study proves that they were more likely to engage with health promotion programmes [38]. As noted in the VALID study, if patients are motivated to attend, then there can be a high uptake of services, as shown by the ≥95% intervention uptake [35]. The evidence supports that this patient subset should be the focus of action-oriented behavioural intervention programmes, including the management of alcohol-use disorders [38].

This study does have limitations. The patient sample may not be representative of all harmful drinkers. Most individuals who attend the community outreach alcohol services have self-presented and often have high levels of dependency; this is likely to have introduced selection bias. The observational design of the study might have also introduced recall bias in the self-reported alcohol intake at index and follow-up visits. To mitigate the risks of recall bias, a structured assessment form specific to services was used. Other limitations include the fact that there was no control group who were not offered transient elastography and that overall, the numbers included were relatively small. The study was conducted at a single centre with a predominantly white ethnic distribution, which thus limits the generalisability of the results.

To our knowledge, this is first study to demonstrate the feasibility of integrating transient elastography into community alcohol services and to explore its role as a biofeedback tool in high-risk drinkers. There is a large burden of undiscovered, asymptomatic, but clinically significant liver disease in patients attending community substance misuse services. By integrating noninvasive liver stiffness testing into these services, an opportunity is created to change the natural history of disease progression in high-risk individuals [16]. The early detection of liver disease followed by targeted interventions is a logical and effective way to reduce the risk of late presentation in liver disease, and to minimise alcohol-related harm. Screening patients with novel biomarkers to demonstrate significant physical damage can have an additional benefit beyond just detecting disease. NIT-based feedback may supplement subsequent decision making in a patients’ behaviour patterns towards a healthier lifestyle [22,39,40,41].

## 5. Conclusions

This study demonstrated that transient elastography is a feasible tool to stratify clinically significant liver disease in a community alcohol service. It significantly enhances the pick-up rate of undiagnosed liver disease among high-risk asymptomatic alcohol users and provides a unique opportunity for early targeted interventions. It can stimulate change in drinking behaviour and can be used as part of the biofeedback to high-risk drinkers. Further research is needed to fully understand the role of NITs as a means of behavioural intervention in alcohol-use disorders.

## Figures and Tables

**Figure 1 biomedicines-10-00477-f001:**
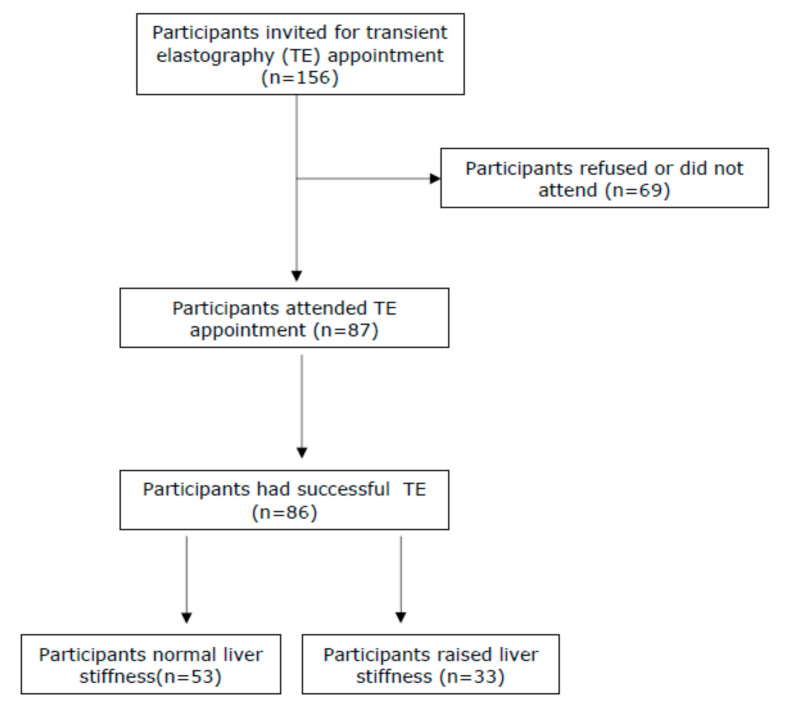
Consort diagram to show participant recruitment.

**Table 1 biomedicines-10-00477-t001:** Characteristics of included cohort.

	Whole Cohort (*n* = 86)	Raised Liver Stiffness (*n* = 33)	Normal Liver Stiffness (*n* = 53)	*p* *
Age (years)	46.3 (±9.8)	46.6 (±8.6)	46.0 (±10.9)	0.79
Gender (male)	53 (70.0)	17 (51.2)	35 (66.0)	
Liver stiffness score (kPa)	6.9 (3.1–75.0)	13.5 (8.1–75)	5.8 (3.1–8)	<0.01
ALT (units/litre)	64.5 (±52.5)	83.1 (±60.8)	53.0 (±43.4)	0.01
GGT (units/litre)	568.6 (±757.4)	1033.6 (±949.7)	226.7 (±260.8)	<0.01
Alcohol intake (units/week)				
Baseline (*n* = 57)	145 (24–420)	149 (39–420)	126 (24–378)	0.338
≥Six months (*n* = 47)	80 (0–315)	65 (0–300)	90.7 (0–315)	

Mean (SD), median (range), number (%). * *p* for significance of difference between normal liver stiffness (<8 kPa) versus raised liver stiffness (>8 kPa) group.

**Table 2 biomedicines-10-00477-t002:** Comparison between normal liver stiffness versus raised liver stiffness subgroups.

	Normal Liver Stiffness	Raised Liver Stiffness	*p*
Alcohol intake ≥ 6 months	*n* = 21	*n* = 26	
Reduced	14 (66.7%)	17 (65.4%)	1.00
Increased	5 (23.8%)	1 (3.8%)	0.08
Abstinence	4 (19.0%)	7 (26.0%)	0.73
Alanine aminotransferase (ALT)	*n* = 53	*n* = 32	0.11
Normal	23 (43.4%)	8 (25.0%)	
Raised	30 (56.6%)	24 (75.0%)	
Gamma-glutamyl transferase (GGT)	*n* = 34	*n* = 25	0.23
Normal	5 (14.7%)	1 (4.0%)	
Raised	29 (85.3%)	24 (96.0%)	

Number (%).

## Data Availability

The data presented in this study are available on request from the corresponding author. The data are not publicly available due to local regulatory restriction on data availability.

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
