# Peer review of "Transient Elastography in Community Alcohol Services: Can It Detect Significant Liver Disease and Impact Drinking Behaviour?"

_biomedicines, 2022, doi:10.3390/biomedicines10020477_

Round 1

Reviewer 1 Report

The topic addressed is interesting, and I have enjoyed reviewing this manuscript. It would be a very good contribution to the journal.

Minor points:

It would be better if the authors could also mention the relationship with Child-Pugh score or malignancy.

Author Response

Comments and Suggestions for Authors

The topic addressed is interesting, and I have enjoyed reviewing this manuscript. It would be a very good contribution to the journal.

Thank you very much for your appreciation. We hope the article will attract the same response from a wider audience and readership.  

Minor points:

It would be better if the authors could also mention the relationship with Child-Pugh score or malignancy.

Thank you very much for your comment. We agree with the reviewer that would have been useful to explore the relationship between Child-Pugh score or malignancy to change in drinking behaviour. Unfortunately, in alcohol services in the United Kingdom, the data to calculate Child-Pugh score and associated malignancies is not routinely collected. Future work in this area could include follow-up of these patients over a substantial time period when the relation of outcomes such as malignancy and stage of liver disease (Child-Pugh, MELD score) could be looked at.

Reviewer 2 Report

 It is a pleasure to review this article. However, there are a few points that need to be addressed. 

1-As you have already mentioned the possibility of selection bias, there is also a possibility of recall bias while reporting the alcohol consumption before and after study and should be noted. 

2- The study lacks a comparison group, and thus it is hard to signify the scientific significance of TE integration in alcohol modification behaviors. 

3- Please state the psychosocial and pharmacological interventions done to reduce the alcohol intake among the patient population. 

4- Was there any reporting of alcohol consumption in between six months duration between two readings?

5- There is a tiny patient population, and it is without comparison. How will you prove the significance of the results for this small population without control? Please explain. 

Author Response

Comments and Suggestions for Authors

It is a pleasure to review this article. However, there are a few points that need to be addressed. 

Thank you very much for taking the time to review our article.

1-As you have already mentioned the possibility of selection bias, there is also a possibility of recall bias while reporting the alcohol consumption before and after study and should be noted. 

Thank you very much for your comment. We have now added recall bias as limitation in the discussion section as suggested by the reviewer.

2- The study lacks a comparison group, and thus it is hard to signify the scientific significance of TE integration in alcohol modification behaviors. 

Thank you very much for your comment. We agree with reviewer and have already included this as a limitation of the study. The study serves as a proof of concept to assess the feasibility of Transient elastography in community alcohol services. We have highlighted this in abstract, discussion and in conclusion sections.

3- Please state the psychosocial and pharmacological interventions done to reduce the alcohol intake among the patient population. 

Thank you very much for your comment. The standard treatment in community alcohol services varies based on individual patient need. In general, the pharmacological interventions include management of alcohol withdrawal and prevention of alcohol relapse. Examples of the most common medications used in these services to prevent relapse are Disulfiram, Naltrexone, and Acamprosate. The psychology interventions comprise of subgroup of interventions such as motivational intervention, family and social network intervention, cognitive behaviour-based relapse prevention, and behavioural self-control.

4- Was there any reporting of alcohol consumption in between six months duration between two readings?

Thank you very much for your comment. The data on self-reported alcohol intake was only collected at baseline and at 6 months follow up.

5- There is a tiny patient population, and it is without comparison. How will you prove the significance of the results for this small population without control? Please explain. 

Thank you very much for your comment. We agree with reviewer comment; the small sample size and lack of control group are limitations. Hence, we have taken the opportunity to present this study as proof-of-concept study to raise an important scientific question which would require definitive trial to prove or disprove the hypothesis specifically related to aspect of NITs as a behavioural intervention in alcohol use disorders.

Round 2

Reviewer 2 Report

na